# RespiCell^TM^: An Innovative Dissolution Apparatus for Inhaled Products

**DOI:** 10.3390/pharmaceutics13101541

**Published:** 2021-09-23

**Authors:** Fabio Sonvico, Veronica Chierici, Giada Varacca, Eride Quarta, Davide D’Angelo, Ben Forbes, Francesca Buttini

**Affiliations:** 1Department of Food and Drug, University of Parma, Parco Area Delle Scienze 27/A, 43124 Parma, Italy; fabio.sonvico@unipr.it (F.S.); giada.varacca@studenti.unipr.it (G.V.); eride.quarta@studenti.unipr.it (E.Q.); davide.dangelo@unipr.it (D.D.); 2Interdepartmental Center for Innovation in Health Products, Biopharmanet Tec, University of Parma, Parco Area Delle Scienze 27/A, 43124 Parma, Italy; veronica.chierici@studenti.unipr.it; 3Institute of Pharmaceutical Science, King’s College London, 150 Stamford Street, London SE1 9NH, UK; ben.forbes@kcl.ac.uk

**Keywords:** RespiCell, dissolution, respirable fraction, pulmonary drug delivery, vertical cell diffusion, dry powder inhaler

## Abstract

To overcome some of the shortfalls of the types of dissolution testing currently used for pulmonary products, a new custom-built dissolution apparatus has been developed. For inhalation products, the main in vitro characterisation required by pharmacopoeias is the deposition of the active pharmaceutical ingredient in an impactor to estimate the dose delivered to the target site, i.e., the lung. Hence, the collection of the respirable dose (<5 µm) also appears to be an essential requirement for the study of the dissolution rate of particles, because it results as being a relevant parameter for the pharmacological action of the powder. In this sense, dissolution studies could become a complementary test to the routine testing of inhaled formulation delivered dose and aerodynamic performance, providing a set of data significant for product quality, efficacy and/or equivalence. In order to achieve the above-mentioned objectives, an innovative dissolution apparatus (RespiCell™) suitable for the dissolution of the respirable fraction of API deposited on the filter of a fast screening impactor (FSI) (but also of the entire formulation if desirable) was designed at the University of Parma and tested. The purpose of the present work was to use the RespiCell dissolution apparatus to compare and discriminate the dissolution behaviour after aerosolisation of various APIs characterised by different physico-chemical properties (hydrophilic/lipophilic) and formulation strategies (excipients, mixing technology).

## 1. Introduction

The in vitro dissolution tests for solid oral dosage forms are well-established characterisation steps necessary to guide formulation and product development as well as to ensure quality control during medicinal product manufacture. This has never been the case for inhalation drug products, as the first and most critical step for their clinical performance involves the delivery of the active pharmaceutical ingredient (API) from the device and its deposition in the nose or lung [1,2]. If the particles are too large or too small, they will not reach or deposit efficiently within the appropriate target organs. This may be a primary reason why in vitro testing of inhaled products focuses on delivered dose and aerodynamic particle size distribution (APSD) and not on drug dissolution.

Drug dissolution, however, is a prerequisite to the absorption or uptake via epithelial cells in the pulmonary tract. For compounds with high water solubility, the dissolution is not a critical step and the therapeutic effect is not influenced significantly by the material or formulation properties. For slow-dissolving compounds, the therapeutic effect is sensitive to material properties governing solubility and/or dissolution rate [3]. Indeed, a direct relationship between the dissolution rate and mucosal permeation has also been reported for inhaled products [4,5].

Until relatively recent times, regulatory agencies’ advisory boards have indicated that dissolution testing did not appear to be a critical characterisation for currently approved inhalation products. In fact, the small dosages of such products, along with reduced particle size, did not suggest any clinical relevance or issues for the dissolution process of APIs in the lung [2]. Nevertheless, the respiratory drug delivery market is moving from low-dose locally acting drugs to higher-dosage, poorly soluble and possibly systemically acting compounds [6]. For this next generation of inhalation products, it appears that the classic pharmacopeial characterisations are no longer appropriate to exhaustively describe the product profile, and the development of more bio-relevant techniques would help to predict the products’ biological availability [7]. Thus, in the future, to an increasing extent, the solid state and dissolution properties of the active compounds will play a crucial role in the pharmacokinetics and pharmacodynamics of inhaled drugs.

Several dissolution techniques for inhaled products have recently been proposed [8,9,10,11] for assessing product quality or bioequivalence. Moreover, a growing interest in using biorelevant dissolution experiments as a tool to better understand the behaviour of inhaled medicines in the lungs has emerged [12,13,14].

The respiratory tract has a large surface area (>100 m^2^) but only a small amount of aqueous liquid (approximately 10–20 mL), and there is the presence of endogenous lung surfactant. Furthermore, there are different clearance mechanisms in the upper conductive and lower respiratory airways. In the upper respiratory tract, a thick film of mucus acts as a physical and chemical barrier, trapping particles and slowing the diffusion of drugs, favouring the mucociliary clearance of the inhaled material. In the lower airways, particles deposited in the alveolar region are exposed to alveolar macrophage phagocytosis; however, the presence of the alveolar surfactant is expected to promote the dissolution even of poorly soluble drug particles [2]. Given all these characteristics, an ideal in vitro dissolution system for inhaled products should not use an excessive volume of dissolution medium, but should still provide sink conditions, reproduce the air-liquid interface in which deposited particles dissolve in the airways and, if possible, mimic lung fluid hydrodynamics. Besides, it should be remembered that only a fraction of the labelled dose is available for local action or systemic absorption. In fact, part of the dose is retained by the inhaler or impacts in the upper airways as a consequence of the large aerodynamic size of some API particles or agglomerates. It follows that a dissolution method for inhaled products should also be able to analyse the respirable fraction of the inhaled dose, excluding from the analysis not only API particles of non-respirable size but also excipients. Indeed, it has been reported that the drug aerodynamic particle size affects drug dissolution [8] and that the dissolution profile of an aerosolised product is different from that of the bulk formulation [15]. Moreover, it has been reported that the dissolution profile is affected by drug dose, the intrinsic API solubility and the drug deposition method in the dissolution apparatus [16].

Despite these difficulties, several in vitro approaches for dissolution testing of inhaled products have been proposed, either by adapting pharmacopoeial dissolution systems or by proposing new apparatuses [4,9,10,17].

For example, USP Apparatus 1 (Basket), 2 (Paddle), 4 (Flow-Through Cell) and 5 (Paddle over Disk), horizontal and vertical diffusion cells and a modified version of the Twin Stage Impinger have been proposed in combination with different types of dissolution media (e.g., phosphate buffer solution, simulated lung fluid or water) [18]. The main shortcomings of these methods are the lack of standardisation regarding the powder deposition method in the dissolution apparatus, the influence on the dissolution of the hydrodynamic conditions applied [19] and the unfeasibility of dissolving only the respirable fraction (below 5 µm) of the formulation [20]. Among the more innovative systems, an interesting apparatus is the PreciseInhale equipment used in conjunction with the DissolvIt cell (Inhalation Sciences, Huddinge, Sweden) [11]. Briefly, the formulation is aerosolised inside a cylindrical holding chamber and deposited by force of gravity on a cover slip. The particles are then transferred to a dissolution chamber, covered with a membrane and perfused with an appropriate medium. The benefits of this system are the simulation of the air-blood interface of the deep lung and the possibility of observing in real time the dissolution of the particles through a microscope. Recently, fluticasone propionate dissolution data obtained by this system were translated by in silico model pharmacokinetics parameters and found to be within 2-fold of the values reported for inhaled fluticasone in humans [21]. However, the limitations of the system are its complex configuration and high cost, as well as the collection of most of the aerosolised powder without a cut-off in the respirable particle size distribution range. Another innovative aerosol dose collection system (UniDose^TM^, NanoPharm Aptar, Newport, UK) was designed to uniformly deposit the whole NGI impactor stage mass on a large filter, then transfer it to a vessel where the velocity of dissolution in 300 mL of medium was measured by an adapted USP paddle over disk apparatus [22]. The validated dose collection method was utilised to demonstrate that the dissolution profiles of both commercial MDI and DPI products were independent of the number of actuations over a wide range of drug loading. The absence of an effect on the dissolution rate as the number of emitted doses increases is certainly due to the collection method, which allows for a homogeneous deposition of the sample on the filter. However, this result is also due to the dissolution system, which involves immersion of the sample and an important hydrodynamic contribution of the agitation due to the paddle.

In order to develop an innovative dissolution system, RespiCell was designed to be intuitive and easy to use and to possess the key characteristics for the discrimination of the release profiles of inhaled products. Furthermore, it can be applied to both soluble and poorly soluble drugs, aerosolised or non-aerosolised (bulk) formulations. Dissolution in RespiCell is carried out by maintaining air-liquid dissolution conditions and using a limited but non-dissolution rate-limiting volume of medium. The apparatus is a customised vertical diffusion cell able to accommodate, as a diffusive membrane, the glass fibre filter collecting the respirable dose of an aerosolised product. The aim was to design an apparatus able to discriminate between the dissolution behaviour of inhalation products and to provide significant data for (i) formulation development, (ii) generic product in vitro equivalence testing and (iii) quality assessment of scale-up or industrial batches.

The hypothesis underlying this work was that the study of the dissolution profiles for a number of different APIs could be helpful for the selection of the more appropriate formulation excipients, the choice of the optimal manufacturing process or the identification of a formulation providing a dissolution profile similar to that of the reference product in order to decrease the risk of failure in bioequivalence pharmacokinetics/pharmacodynamics (PK/PD) studies.

In particular, in this paper, the RespiCell apparatus was applied to assess the impact on the dissolution kinetics of magnesium stearate in tobramycin powders and of different blending processes in mixtures of two different new chemical entities (NCEs) with lactose. Furthermore, various tiotropium or indacaterol formulation dissolution profiles were compared with those of their respective reference products. Finally, how the production scale-up from laboratory scale to semi-industrial GMP batch impacted on the release profiles of selected tiotropium formulations was determined.

## 2. Materials and Methods

### 2.1. Materials

Tobramycin was purchased from TEVA, and magnesium stearate (MgSt) was supplied by Peter Greven (Bad Münstereifel, Germany). Raw materials of NCE-A and NCE-B were synthesised and micronised by the University of Parma (Parma, Italy). Lactohale 200 (DFE Pharma, Goch, Germany) was used to prepare NCE-A and NCE-B powder blends.

Tiotropium bromide (TBr) micronised raw material, Spiriva HandiHaler^®^ 18 µg/5.5 mg (batch numbers: 703876B; 702567; 706595; Boehringer Ingelheim, Germany) and TBr test formulations prepared using lactose monohydrate as carrier were provided by Delim Cosmetics & Pharma S.r.l. (Milan, Italy). Indacaterol maleate (IDC) micronised raw material, Onbrez Breezhaler^®^ 150 µg (batch numbers: BCF19, BU725, BT617; Novartis, Switzerland) test formulations were supplied by Delim Cosmetics & Pharma Srl. In these mixtures, micronised IDC was blended by high shear mixing with coarse lactose and increasing proportions of fine lactose (5, 10 or 15% *w*/*w*). Both fine (Lactohale LH300) and coarse (Respitose ML001) lactose monohydrate were supplied by DFE Pharma (Goch, Germany). The aqueous solubilities of all the drugs used are reported in Appendix A.

HPMC Quali-V-I size 3 capsules were provided by Qualicaps (Madrid, Spain) and gelatin V-Caps Plus by Capsugel^®^ (Colmar, France). A single dose high resistance dry powder inhaler RS01_HR was a gift from Plastiape (Osnago, Italy).

Simulated lung fluid (SLF) was prepared as described previously, and its composition is reported in Appendix A. In specific cases, sodium dodecyl sulphate (SDS, Merk, Darmstadt, Germany) or Tween 80 (Croda Italiana, Mortara, Italy) were added to PBS or SLF as surfactants.

For analytical and preparative purposes, the following reagents were used: phosphoric acid, 85%, sodium hydroxide, potassium dihydrogen phosphate (A.C.E.F. S.p.A., Fiorenzuola d’Arda, Italy), sulphuric acid, sodium dihydrogen phosphate, tris(hydroxymethyl) aminomethane (Merk, Darmstadt, Germany) and ethylenediaminetetraacetic acid (EDTA; Riedel-de Haen; Germany). All solvents used were of analytical grade and water was ultrapure (resistivity = 1–10 MΩ·cm, conductivity = 1–0.1 µS/cm; Purelab Flex, ELGA-Veolia LabWater, High Wycombe, UK).

### 2.2. High Performance Liquid Chromatography (HPLC)

The drugs were analysed by means of a validated HPLC method using an Agilent 1100 Infinity LC system (Agilent Technologies, Santa Clara, CA, USA) equipped with a UV–Vis detector, auto-sampler, degassing unit and column oven and operated with OpenLab CVS Chem Station software (rev.c.01.06 v.A.04.02, Agilent Technologies). A specific HPLC method was validated for each individual drug. The analytical methods were validated for linearity, precision and accuracy in the appropriate concentration range of analysis in accordance with ICH Guideline Q2 (R1). The details and description of HPLC method parameters are reported in Appendix A.

### 2.3. Dry powder Formulation Preparation Processes

Two different new chemical entity micronised powders, NCE-A and NCE-B, were mixed with lactose carrier at different strengths with different blending approaches: Turbula™ Mixer (Willy A.Bachofen AG; Nidderau-Heldenbergen, Germany), MINI Cyclomix (Hosokawa Micron B.V.; Doetinchem, The Netherlands) and Resodyn Acoustic Mixer (LabRAM, Resodyn Corporation, Butte, MT, USA). In this way, 12 blends were obtained. The uniformity of drug content in the blend was determined by HPLC (*n* = 6). Blends were considered homogeneous with a precision of API content of ±5% compared to the target dose value and within an accuracy in terms of relative standard deviation of <5%. Tobramycin was blended with MgSt at two different concentrations of 0.5% and 5% *w*/*w* using a high shear mixer (MINI Cyclomix) with a mixing time of 10 min and speed at 3000 rpm. The batch size was 20 g of powder, and the smallest vessel of 0.1 L was employed.

TBr formulations were prepared by blending micronised TBr and lactose in the ratio of 22.5 µg of drug (equivalent to 18 µg of base) in 5.5 mg of powder. A high-shear process was employed to prepare pilot and GMP batches by Chance Pharmaceuticals (Chance Pharmaceuticals Co. Ltd., Hangzhou, China). Test formulations differ for the process parameters employed.

The blending of IDC and carrier was performed using a TriChop Mixer (Lleal, Spain) in order to obtain 194.4 µg indacaterol maleate (equal to 150 µg of indacaterol free base) in 25 mg of blend. For the carrier preparation, the fine lactose was mixed with coarse lactose for 5 min at 45 rpm then for another 5 min at 100 rpm. The API was then placed between two portions of the prepared carrier and mixed for 10 min at 45 rpm, then for another 10 min at 150 rpm. In both processes, the vacuum was set at −0.25 bars and air pressure from diffusers at 0.2 bars.

### 2.4. Dissolution Studies

#### 2.4.1. Dissolution Apparatus and General Procedure Description

In vitro dissolution studies were conducted using RespiCell™ (EU registration No. 006649570-0001, designed by the University of Parma and produced by DISA s.p.a., Milan, Italy), a vertical diffusion cell apparatus specifically designed for the dissolution test of inhaled products (Figure 1). The apparatus consists of two portions: the upper part acts as a donor compartment and the lower part is the receptor compartment. The receptor chamber is filled with 170 mL of dissolution medium magnetically stirred (magnet of 2 cm length) and has a sampling side arm of 10 cm length. The two compartments are held together by a metal clamp, and they are separated by a glass fibre filter, which is used as diffusion membrane as it sits directly in contact with the dissolution medium. The diffusion area is approximately 30.2 cm^2^.

The dissolution conditions selected for each product were those best able to distinguish the differences between the different formulations or products. In particular, the dissolution medium was selected according to the intrinsic water solubility of the API. In fact, for poorly soluble molecules an appropriate surfactant was added to the dissolution medium in order to assure sink conditions (API final concentration below 10% of its solubility) [23]. Type A/E glass filters of 7.6 cm diameter, (PALL Corporation, Port Washington, NY, USA) were employed in all these experiments as diffusion membranes. The receptor was filled with 170 mL of dissolution medium, which was sampled at a predetermined time through the side arm of the cell, according to the requirements of the experiment. At fixed intervals, 1 mL of the receiving solution was collected and replaced with 1mL of fresh medium. RespiCell was connected to a heating thermostat (Eco Silver E4, Lauda, Assago, Italy) set at 37 ± 0.5 °C. After powder sample deposition on the filter, 2 mL of dissolution medium was added to the donor compartment in each experiment. At the end of each experiment, the residual powder was also recovered by washing out the filter with 10 mL of medium in order to assess the amount of drug not dissolved and/or entrapped in the filter. Each blend was analysed in at least three independent experiments and data were expressed as percentage of drug dissolved. The total amount of drug for the dissolution experiment (corresponding to 100%) corresponded to the amount of drug dissolved plus the amount recovered from the donor compartment and in the filter. In any case, for all the experiments the total mass recovered at the end of each experiment by HPLC analysis was >90% of the drug amount deposited.

#### 2.4.2. Collection of Fine Respirable Fraction for Dissolution Profile Test

The dissolution test was performed, when not indicated otherwise, after powder aerosolisation. The respirable particle fraction (aerodynamic particle size below 5 µm) was collected using a fast screening impactor (FSI, Copley scientific Ltd., Nottingham, UK). This impactor allows us to divide the aerosol into two fractions: the first one is composed of particles with an aerodynamic diameter >5 µm, collected inside the induction port and the coarse fraction collector (CFC), and the second consisting of particles with an aerodynamic diameter <5 µm, collected in the fine fraction collector (FFC) on a glass fibre filter. The different inhalation powders under investigation were aerosolised using different types of inhalers, as detailed in Section 2.4.3. However, for each experiment, the airflow rate was set in order to have a 4 ± 0.8 kPa pressure drop, and the FSI was equipped with the appropriate CFC insert according to the airflow rate applied. The sampling time was set by letting a volume of 4 litres pass through the system. In all the experiments, the emitted dose (ED) was defined as the quantity of drug deposited in the FSI and quantified by HPLC, the respirable dose (RD) was the mass of the drug with aerodynamic diameter lower than 5 µm and the respirable fraction (RF) was calculated as the ratio of the RD to the ED plus the amount of API recovered from the capsule and inhaler and quantified by HPLC.

#### 2.4.3. Specific Details Adopted for the Dissolution Test of Different APIs

Tobramycin blends were analysed without aerosolisation; 200 mg of powder were deposited on the wet filter, and ultrapure water was used as dissolution medium.

NCE blends were aerosolised using an RS01 device operated with an airflow of 60 L/min. Capsules were filled with 20 mg of powder, and a different number of capsules was aerosolised depending on the API strength and respirability to obtain approximately the same mass of drug deposited on the filter (1 mg for NCE-A and 0.9 mg for NCE-B). PBS (pH 7.4) containing SDS 0.5% *w*/*v* was used as a dissolution medium for both the molecules.

Spiriva and TBr test formulations were aerosolised inside the FSI using the HandiHaler device at 45 L/min. The content of 10 capsules was discharged into the impactor. The RF was of 23.5% for Spiriva and between 20–23% for the three test batches. In this case, PBS pH 6.8 with EDTA 0.025% *w/v* was used as a dissolution medium.

Onbrez and IDC test formulation were aerosolised inside the FSI using the Breezhaler device at 90 L/min. The content of one capsule having a dose of 150 µg indacaterol as a free base was discharged into the impactor. The RF was in the range 49–52% for both Onbrez and the test formulation. SLF (pH 7.4) with 0.2% *w*/*v* of Tween 80 was used as a dissolution medium.

#### 2.4.4. Difference and Similarity Factors between In Vitro Dissolution Profiles

The dissolution profiles were examined in terms of both fraction and overall amount dissolved over time, using the difference (𝑓1) and similarity factors (𝑓2) already proposed to compare the dissolution profiles of oral dosage forms [24,25]. The difference factor (𝑓1) calculates the percent difference between the two dissolution profiles at each time point and is a measurement of the relative error between the two profiles:(1)$$f_{1}=\left(\frac{\sum_{t=1}^{n}\left|R_{t}-T_{t}\right|}{\sum_{t=1}^{n}R_{t}}\right)\times 100$$

The similarity factor (𝑓2) is calculated as follows:(2)$$f_{2}=50\times \log\left[\frac{100}{\sqrt{1+\frac{\sum_{t=1}^{n}{\left(R_{t}-T_{t}\right)}^{2}}{n}}}\right]$$
where *n* is the number of time points, *R_t_* is the mean dissolution value for the reference product at time *t*, and *T_t_* is the mean dissolution value for the test product at that same time point. For both the reference and test formulations, percent dissolution should be determined. The evaluation of the similarity factor is based on the following conditions: a minimum of three time points (zero excluded) should be considered, and the time points should be the same for the two formulations, and not more than one mean value should exceed 85% of the dissolved drug for any of the formulations. In addition, the relative standard deviation (coefficient of variation) should be less than 20% for the first time point and less than 10% for the other time points considered. A difference factor (𝑓1) value lower than 15 (0–15) indicates no significant difference between the dissolution profiles. A similarity factor (𝑓2) value higher than 50 (50–100) indicates similarity between two dissolution profiles.

### 2.5. Scanning Electron Microscopy (SEM)

The morphology powders were determined using scanning electron microscopy (SEM, Zeiss AURIGA, Oberkochen, Germany) operating under high vacuum conditions with an accelerating 1.0 kV voltage, at different magnifications. Powders were deposited on adhesive black carbon tabs pre-mounted on aluminium stubs and imaged without undergoing any metallisation process. When only the fine fraction morphology was analysed, the sample was collected by placing double-sided carbon tape adhesive on the FFC filter of the impactor before aerosolisation. The tape was then mounted on the aluminium stub.

### 2.6. Aerodynamic Assessment by Next Generation Impactor of Indacaterol Maleate

The aerodynamic performance of IDC batches was assessed using the NGI (Copley Scientific, Nottingham, UK). The NGI was connected to a SCP5 vacuum pump (Copley Scientific, UK) through a TPK. The cups of the impactor were coated with a thin layer of ethanol containing 2% (*w*/*v*) Tween 20 to prevent particle bounce. The API found in the induction port, in the pre-separator, in the different stages and in the MOC of the NGI was collected using water: acetonitrile (70:30 *v*/*v*); the residual API in the device and in the capsule was collected with water: methanol (70:30 *v*/*v*) and assayed by HPLC as previously reported. Owing to the resistance of the Breezhaler^®^ device, the in vitro aerodynamic assessment of this IDC was tested at 100 L/min air flow rate in order to achieve a 4 kPa pressure drop. The flow rate was set using a DFM 2000 Flow Meter (Copley Scientific, Nottingham, UK). The content of one capsule of three batches containing different amounts of fine lactose (IDC_Fine 5%, IDC_Fine 10%, IDC_Fine 15%) was aerosolised for each NGI test and compared with Onbrez. The aerodynamic parameters were calculated according to Ph.Eur. 10th Edition.

## 3. Results and Discussion

The study of the solid state of the API, of its modification induced by the processing and of the resulting dissolution properties is fundamental for the development of innovative lung products, as well as for the development of bioequivalent drugs. Hence, the time has come to design several novel experimental approaches specifically tailored to evaluate the unique characteristics of lung-directed formulations. These new techniques, once validated and implemented, have to produce robust data to complement traditional aerosolisation performance studies that seem inadequate for the challenges posed by the rapidly changing landscape of the inhalation product market.

In this respect, the RespiCell dissolution apparatus has been designed with the aim of having a system able to offer a liquid-air interface for the dissolution and a reduced receiver volume compared to that of the dissolution apparatuses using vessels, such as those adapted from USP. Furthermore, the diffusion area was decided as being equal to that on which the fine portion of an aerosol is collected by impactor analysis. This feature allows, if desired, for monitoring the dissolution profile only of the respirable fraction of a formulation by simply mounting on the diffusion cell the filter collecting the fine particle fraction from the impactor after inhalation product aerosolisation.

As with all the vertical diffusion cells, the RespiCell apparatus features the limitation that the profiles may be dependent on the quantity of sample subjected to analysis, especially for poorly soluble compounds analysed in bulk, for which the penetration of the fluid into the powder bed is favoured by small and evenly spread amounts of formulation. In this context, a standardised method for powder loading on the membrane is highly useful for data robustness. This aspect appears less critical if the product is collected after aerosolisation inside an impactor. Modified versions of the Andersen cascade impactor have been tested to increase the homogeneity of the particle distribution [26]. In our case, the collection was made using the fast screening impactor where the product appeared distributed in clusters arranged in an orderly manner over the glass fibre filter present in the fine fraction collector.

The results of the studies reported below highlight the discriminatory performance of the RespiCell apparatus, even in unfavourable conditions, such as highly and poorly soluble drugs, identical formulations mixed with different techniques, formulations of high or low strength dose active ingredients and differing batches of generic formulations.

### 3.1. Dissolution Test as a Tool to Investigate the Impact of Formulation Composition on Drug Release

The first set of data presented illustrates how, by using RespiCell, it is possible to appreciate the differences in drug release profiles induced by modifying the composition of the formulation. This aspect is particularly useful at the initial stages of formulation design.

Here, as an example, we present the investigation of tobramycin-based powders coated with magnesium stearate (MgSt) with the aim of slowing down the dissolution profile of the drug. Tobramycin, as a free base, is an aminoglycoside antibiotic freely soluble in water (1000 mg/mL) used for the local treatment of lung infection in patients with cystic fibrosis. The rationale for decreasing the release rate of the drug after deposition in the airways was to sustain tobramycin local concentrations at the infection sites above its minimum inhibitory concentration (MIC) for longer periods. The results demonstrate that the addition of increasing proportions of MgSt (0.5–5% *w*/*w*) slowed the tobramycin dissolution rate compared to that of the reference formulation, for which almost 90% of the drug dissolved after only 2 min. In particular, the powder containing 5% *w*/*w* of MgSt showed a significantly slower profile than the powder with 0.5% (Figure 2), as confirmed by the results of difference (𝑓1) and similarity (𝑓2) factors (52.19 for 𝑓1 and 20.47 for 𝑓2). At the end of the experiment, both formulations released >96% of the drug deposited as bulk formulation on the glass fibre membrane, while the small residue still present in the donor was attributed to the presence of the insoluble MgSt.

### 3.2. Effect of Blending Process on Dissolution of a Poorly Soluble API

The second example of application we report is the study of how the mixing process of a carrier lactose and a micronised API could impact the dissolution profile of the drug. In detail, the dissolution behaviour of the respirable fraction of two different poorly soluble molecules (NCE-A and NCE-B) mixed with a coarse lactose (90–150 µm) was assessed. Blends were prepared by means of the classic low shear process (Turbula), using a high shear impact mixer (Cyclomix) or employing an acoustic mixer (Resodyn Acoustic Mixer). The high shear impact mixer has a central rotor that pushes particles towards the conic vessel wall, while the acoustic mixer uses low frequency high intensity acoustic energy to blend the powder bed. The latter two alternative methods consume much less time than Turbula to obtain an ordinate mixture.

NCE-A was the first molecule included in the study and it was mixed with lactose at 5% *w*/*w* with all three apparatuses. The mixtures were found to be homogeneous with all processes (accuracy ± 5% of API as compared with the expected content and precision <5%). Even from the point of view of aerodynamic behaviour, no substantial differences were highlighted using the fast screening impactor. In all cases, the emitted dose was greater than 80%, and the respirable fraction was around 50%. From these results and from the need to deposit several doses on the filter to obtain an analytically appreciable mass of API, it was set that about 1 mg of drug was needed for the dissolution study. Moreover, a preliminary work was necessary to select the surfactant and its concentration to use in the dissolution medium. It is in fact important to find a compromise between a quantity of surfactant sufficient to assure dissolution in sink conditions but on the other hand not so high as to make it difficult to discriminate the differences between the different formulations. In this specific case, PBS with 0.5% SDS was selected.

The profiles of the three NCE-A mixtures (Figure 3A) showed that, in all cases, 80% of the drug dissolved within 3 h. However, slight differences between the profiles were statistically analysed taking the Turbula mixed batches as references to compare profiles of Cyclomix and acoustic mixer.

The results obtained by difference and similarity factors showed no difference and similarity if the dissolution profile of the blend obtained with Turbula was compared with the one obtained with the acoustic mixer. However, if the dissolution profiles of the Turbula and Cyclomix blends were compared, the dissolution curves appeared no different in terms of shape (𝑓1 < 15), but at the same time the curves were not similar: this is demonstrated by the fact that the curves crossed and that the value of 𝑓2 factor was <50 (48.9). In summary, the three different mixing techniques did not excessively affect the physico-chemical properties of the NCE-A, which consequently behaves in the same way for all the blends.

The second poorly soluble molecule included in this study was NCE-B, mixed at 4% *w*/*w* with the carrier by Turbula, Cyclomix or Resodyn Acoustic. All three blending processes led to mixtures that were within the limits of accuracy (±5%) and precision (CV < 5%), and PBS with 0.5% SDS was selected again as the most suitable dissolution medium.

Unlike NCE-A, NCE-B blends had differing respirability, indicating that the manufacturing process affected this product’s aerosolisation performance. Although, in all cases, the ED was above 80% and the amount deposited in the induction port of FSI was around 5%, the RF was around 57% for blends prepared with the Turbula or the acoustic mixer, but 40% for those prepared using Cyclomix. From this finding and from the need to dissolve similar masses of active ingredients, an extra dose was aerosolised for this last formulation in order to have, in all cases, about 0.9 mg of NCE-B deposited on the filter. The dissolution of the fine fraction was then assessed.

Regarding the dissolution profiles (Figure 3B), those of the Turbula and acoustic mixer blends were quite similar, while that of the Cyclomix powder was distinctly faster, especially in the first hour. The results obtained calculating the difference and similarity factors comparing the Turbula and Cyclomix blends showed a difference (𝑓1 = 111.2), but there was no similarity between the two dissolution profiles (𝑓2 = 26.5). In contrast, the blend obtained with the acoustic mixer, compared to the reference Turbula, showed no difference (𝑓1 = 5.9) or similarity in the dissolution profiles (𝑓2 = 82.1). The differences evidenced led to 80% of dissolved API for the Cyclomix blend after 3 h of the experiment, while only 40% of the drug was dissolved for blends obtained with the other two mixing processes.

In order to find an explanation for this significantly different behaviour, a morphological analysis was performed post-aerosolisation (Figure 4). Quite interestingly, a different morphology between the Cyclomix and the other two blends was evidenced in the SEM images. The powder processed with Cyclomix featured the formation of extremely small needle-like crystals. It is still not clear whether these are individual structures or surface modifications of the original crystals. As previously mentioned, Cyclomix is a high energy process that applies forces of impact and friction potentially able to modify the solid state of particles. Therefore, it is likely that the high mixing speed provided sufficient energy to induce a process of surface melting and re-crystallization of the API crystals, leading to the formation of smaller-particle needles, resulting in a faster dissolution, possibly as a result of the increased surface area (Figure 4). Interestingly, the Cyclomix process has been proposed as a dry coating process, designed to mechanically fuse additives and active materials together [27]. This modification applied on a carrier particle surface was claimed to alter morphological properties and particle interactions and in that specific case to improve aerosol performance. Low energy blending (Turbula and Resodyn Acoustic Mixer) did not modify the crystals, which maintained their original shape and size. This important morphological modification caused by high shear impact mixing has been related to the specific characteristics of the NCE-B drug, because it was not observed for NCE-A nor in a previous work focusing on salmeterol xinafoate [15]. 

The surface modification observed was also linked to the low respirability observed for this NCE-B blend, as it was hypothesised that the high shear mixer processing also resulted in an increased adhesion of the active ingredients to the lactose carrier. This was supported by the higher API mass collected in the coarse fraction collector of the impactor.

This part of the work highlighted the ability of RespiCell to discriminate products identical from the point of view of composition but processed in a different way. In this sense, the air-liquid interface, as well as gentler hydrodynamic conditions, were essential because a paddle over disk method would probably have led to a faster dissolution less likely to discriminate the products.

Moreover, the RespiCell proved to be extremely convenient for the analysis of the respirable fraction. In fact, the whole fraction of particles below 5 µm was collected on a single filter and easily transferred to the dissolution apparatus. It has been reported that there may be differences in the dissolution profile between powders before or after aerosolisation [15], and this makes this apparatus relevant for both these conditions.

Finally, even more importantly, the dissolution analysis highlighted physico-chemical differences between the formulations that would not have been highlighted by carrying out only the pharmacopeial aerosol performance test focusing on the APSD. The evidence that a mixing process can affect the release profile on an API in vitro could be of extreme relevance for the possible implications for PK/PD studies.

### 3.3. Dissolution Test to Reinforce the Equivalence between a Generic and a Reference Product

This section aims to illustrate the application of RespiCell to the development of generic or hybrid inhaled products, merely to point out the fact that generic products demonstrate bioequivalence to the reference product by means of PK bioavailability studies while hybrid products demonstrate equivalence by means of pharmacodynamic PD or clinical endpoints.

The development of these products in the inhalation field is increasing, both because the patent coverage of some products has expired and because the incidence of respiratory diseases is constantly growing. The overall systems for bringing generic orally inhaled drugs to the market bear some similarities across countries, but they also display important differences due to the unique cultural, historical and economic circumstances of each region. Even between the United States and Europe, scientific challenges and differences in approaches exist within and between regulatory agencies: while the FDA requests an aggregate weight-of-evidence approach, which utilizes in vitro, PK and PD studies to establish bioequivalence of inhalation products, the European Union’s stepwise approach allows for approval based only on the PD or clinical endpoint equivalence if the in vitro and/or PK studies fail to show equivalence. However, whatever the path to be taken, in vitro data concerning the similarity of the dissolution profile between the reference and the generic product can be of help to reduce the risk of failure of PK/PD studies. This aspect, which is already useful for more water-soluble drugs, becomes even more relevant for poorly soluble active ingredients, for which slight changes in the physico-chemical characteristics can lead to significant differences in the rate of dissolution and therefore in their pharmacological action.

Tiotropium bromide was chosen as an example of a water-soluble molecule for the development of a generic product of Spiriva. Three batch tests were produced starting with the same raw materials but prepared with different parameters of the high shear mixing process. The three test formulations, when compared with Spiriva, showed an equivalent emitted dose of 10 ± 0.3 µg, fine particle mass of 3.0 ± 0.2 µg and similar MMAD around 3.2 µm when aerosolised using the Handihaler device. For the dissolution studies, the respirable mass of ten capsules was collected and the dissolution behaviour analysed using PBS (pH 6.8) with EDTA 0.025% *w*/*v*.

Owing to its high aqueous solubility (25 mg/mL), 80% of the tiotropium dissolved within 15 min in all the experiments (Figure 5). At the end of the dissolution, in all cases the percentage of tiotropium bromide recovered in the donor and trapped in the filter was not more than 7%. Statistical analysis made it possible to determine that only one test formulation (TBr Batch #C) had a dissolution profile statistically similar to that of Spiriva (the difference factor was 10.5 and the similarity factor was 60.9).

From the SEM images of the tiotropium bromide blends investigated, no evident differences in the morphology of the particles resulting from the high shear process were observed. However, mixing conditions with lower energy input (TBr batch #C) resulted in a slower release of the drug. In this regard, it has already been pointed out that the higher the energy applied in the process, the more efficient is the dispersion of active aggregates with a consequent increase in the rate of dissolution. If this effect was particularly evident for salmeterol xinafoate before aerosolisation [15], in this case it was also observed after aerosolisation. Observing the SEM images from the figure of batch #C, it seems that in fact there are clusters of not completely de-aggregated particles, not present in batches #A and #B, where particles appeared to be evenly distributed and more disaggregated (Figure 6).

Indacaterol maleate (IDC) was chosen as an example of a poorly water-soluble (0.8 µg/mL) drug for the development of a generic product of Onbrez Breezhaler. In this case, various percentages of fine lactose were added to the formulation in order to obtain an APSD comparable to that of the reference product. Mixtures based on micronised IDC, coarse and fine lactose at 5, 10 or 15% w/w were prepared by high shear mixer and analysed. All of them were homogeneous (CV < 5%) and with an IDC content of 150 µg ± 5% per capsule. The amount of API retained in the capsule and inhaler after aerosolisation was higher for the test formulations than for the reference (Table 1). From the aerodynamic deposition obtained using a next generation impactor, it was possible to observe that the batch with the 5% of fine lactose had a profile closer to that of the reference product. Equivalent emitted doses as well as mass deposited in the induction port and pre-separator were collected for both the reference and this specific test product (Table 1 and Figure 7). However, analysing in detail the distribution on the different stages of the NGI for the Onbrez and the blend containing 5% of lactose fines, the reference product showed a higher fine fraction at stages (S) 3, 4 and 5 (Figure 7). This difference was outside the in vitro equivalence EU acceptance range of ±15%. In this regard, it is necessary to specify that a robust in vitro deposition equivalence evaluation should be carried out on at least three different batches of the reference product and of the test product to compensate for the high inter-batch variability of the inhalation product [28].

Despite these APSD differences, the dissolution profile of the formulation with 5% of fine lactose (IDC_Fine 5%) was analysed and compared with that of the reference product. As for TBr, also in this case the dissolution was carried out only on the fine fraction collected after aerosolisation of one capsule. The selected dissolution medium was in this case SLF with the addition of Tween 80 at 0.2% *w*/*v*.

The dissolution profile of IDC was very slow, owing to the poor solubility of the drug in aqueous media (Figure 8). Solubility was further slowed because the drug is in its ester form. However, in both cases, after 5 h more than 96% of the drug deposited on the membrane was dissolved, and the quantities left in the donor and in the filter were negligible. Statistical analysis made it possible to determine that the test formulation had a dissolution profile similar to and superimposable on that of the commercial product, Onbrez (the difference factor was 5.03 and the similarity factor 72.14).

The test was particularly useful in this case because it showed that, although the two products did not have an identical aerodynamic distribution, this did not involve a modification of the dissolution kinetics in vitro. This result shows that the contribution to the dissolution profile of the higher fine fraction of the reference product was negligible and did not change the release towards a faster profile. In this respect, it has been previously demonstrated that the dissolution rates of low solubility smaller particles are poorly influenced by the amount of loading drug [8].

If these data are transferred to the context of a generic product development path, they indicate that a PK/PD similarity may exist, even in the absence of a perfect overlap of the APSD, and this may be a support for proceeding with the efficacy studies.

The study of the dissolution behaviour with RespiCell related to the generic drug development approach has led to interesting considerations. The dissolution study has shown that there may be differences for formulations with the same aerodynamic profile and that there may be similar release profiles for formulations with APSD that cannot be perfectly superimposed.

The RespiCell system, combined with an accurate setting of experimental conditions, demonstrated the ability to highlight differences even in the case of Tbr, where discrimination was difficult owing to the high solubility of the molecule.

The in vitro release study represents an additional tool to demonstrate the similarity between two products and to identify the best candidate formulation to undergo subsequent equivalence studies using PK/PD endpoints. In particular, being able to follow the release profile of the whole respirable fraction of the drug (<5 µm) is particularly useful compared to other dissolution systems that offer the collection and analysis only of specific and limited dimensional fractions that do not represent the entire respirable portion of the product.

### 3.4. Industrialisation: Dissolution of Big Size GMP Batch vs. Pilot Lab Batch

Finally, RespiCell can be used routinely in the quality control of GMP batches of inhaled products. In particular, this analysis could be added to the standard controls in order to monitor another product critical quality attribute. The reproducibility of the dissolution profile would further support the robustness of the manufacturing process by ensuring the similarity of GMP batches dedicated to bioequivalence studies.

The batch of TBr-lactose blend, which on a laboratory scale had shown aerodynamic deposition and dissolution characteristics similar to those of the Spiriva product, was then produced on a small industrial scale by high-shear mixer under GMP conditions. Three batches of this formulation were produced and analysed. The three GMP blends presented accurate uniformity of drug distribution and the same deposition upon aerosolisation with an emitted dose of 10.5 µg and a respirable dose in the range 3.2–3.4 µg.

Dissolution tests were performed in duplicate for the three GMP batches, and all single runs are shown in Figure 9. As above, only the respirable fraction was tested, and PBS with 0.25% EDTA was used as dissolution medium.

The release profiles obtained clearly show a fast release of the drug: at the third minute, 70% of the drug was dissolved, and after 15 min more than 90% was dissolved. At the end of the test, the amount of drug dissolved by the three batches (3.2 ± 0.22 µg) was superimposable. The profiles obtained from the three GMP batches were identical to each other, indicating that the production process was reproducible, providing formulations with similar aerodynamic behaviour. The CV calculated for all the time points was always <5%. RespiCell also proved to be a robust and precise system for routine quality control analysis of inhalation products.

## 4. Conclusions

The present work showed that RespiCell, used with appropriate experimental settings, is able to provide useful elements for the development of inhalation formulations as well as for the identification of suitable blending processes. The system was also helpful in discriminating generic formulations and identifying those with a dissolution profile more similar to that of the reference products. Finally, it contributed to the confirmation that the drug release was not altered by an increase in the scale of the production process. Overall, the RespiCell dissolution apparatus has proven to be a robust tool for the development and quality control of inhalation products.

In the future, it is expected that the use of RespiCell with more sophisticated and bio-relevant dissolution media containing mucus, or by using media simulating the differences between the conductive and respiratory airways, could allow for the use of the obtained data to achieve a biowaiver. In fact, a robust and validated method to study the dissolution of inhalation products and providing relevant information for their in vivo behaviour could represent a pivotal step for some regulatory bodies regarding clinical studies for the approval of generic products.

## Figures and Tables

**Figure 1 pharmaceutics-13-01541-f001:**
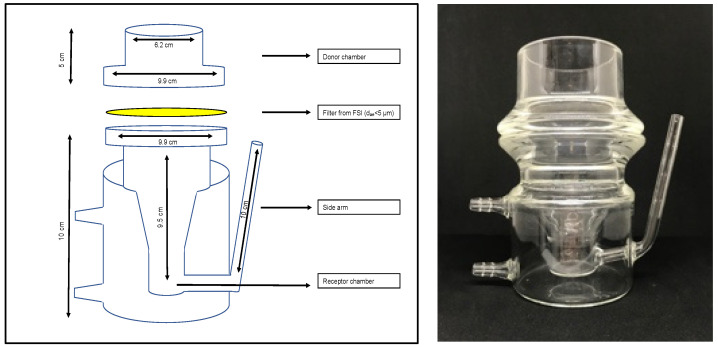
From the left, schematic representation, measurements and pictures of RespiCell™ apparatus.

**Figure 2 pharmaceutics-13-01541-f002:**
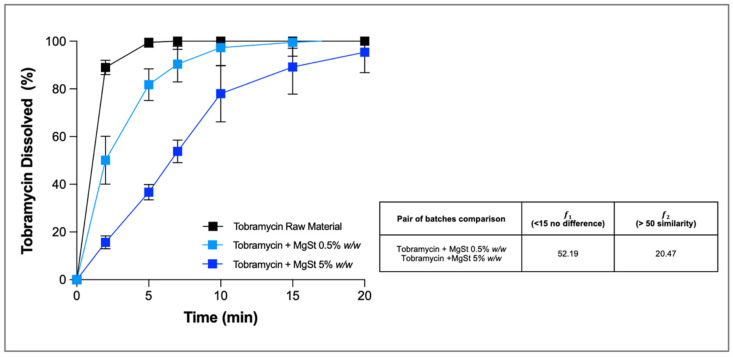
Cumulative mass (%) dissolution profiles of tobramycin bulk inhalation powders containing increasing amounts of magnesium stearate (0% *w*/*w* black, 0.5 *w*/*w* light blue and 5% *w*/*w* blue symbols) (*n* = 3, mean ± SD).

**Figure 3 pharmaceutics-13-01541-f003:**
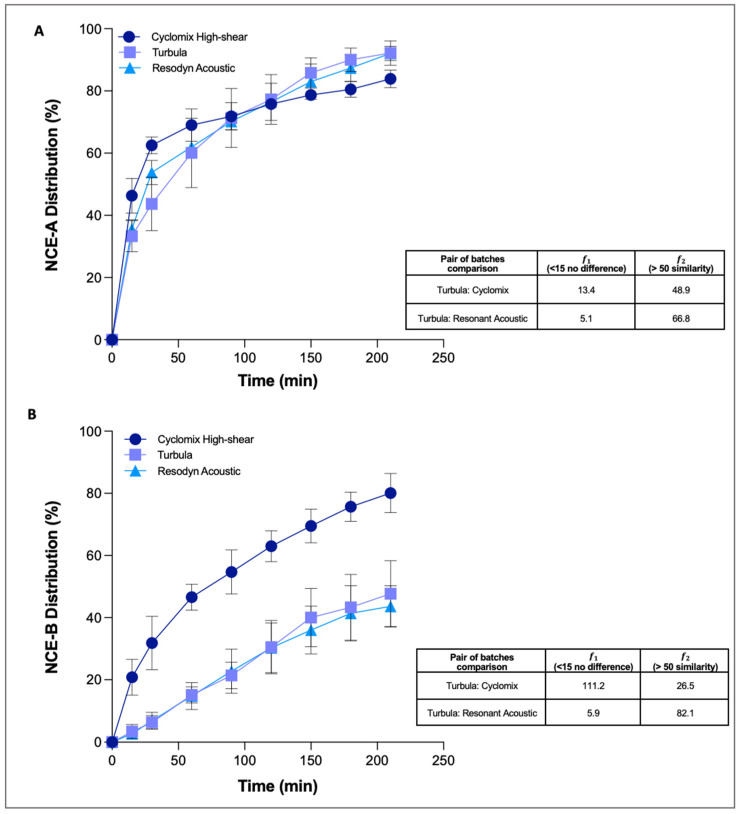
Cumulative mass (%) dissolution profiles of lactose blend formulations containing NCE-A (at 5% *w*/*w*) (**A**) and NCE-B (at 4% *w/w*) (**B**) after aerosolisation using RS01 inhaler (*n* = 3, mean ± SD). Difference (𝑓1) and similarity factor (𝑓2) were calculated taking the batches mixed using Turbula as a reference.

**Figure 4 pharmaceutics-13-01541-f004:**
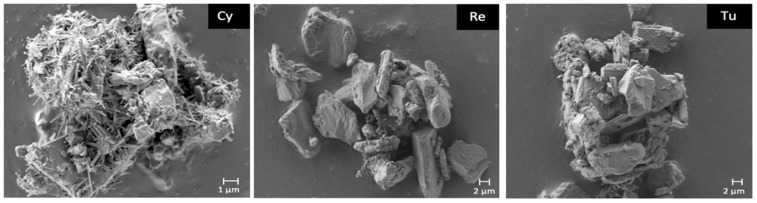
Scanning electron microscopy images of NCE-B blends obtained with Cyclomix high shear impact mixer (Cy), Resodyn Acoustic Mixer (Re) and Turbula (Tu) after aerosolisation (magnification ×20 K). Particles were retrieved in the respirable dose collector of the fast screening impactor.

**Figure 5 pharmaceutics-13-01541-f005:**
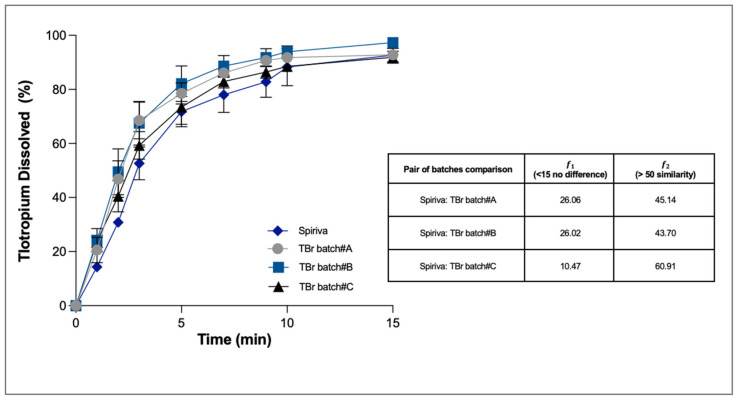
Cumulative mass (%) dissolution profiles of reference product Spiriva (18 µg blended in 5.5 mg of powder) and tiotropium test formulation blends after aerosolisation (*n* = 3, mean ± SD). Difference (𝑓1) and similarity factor (𝑓2) were calculated taking the Spiriva as reference.

**Figure 6 pharmaceutics-13-01541-f006:**
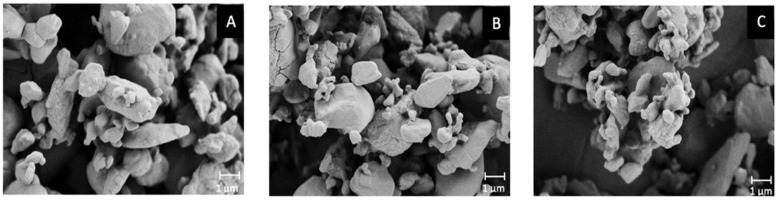
Scanning electron microscopy images of tiotropium test blend formulations: (**A**) Tbr batch #A; (**B**) Tbr batch #B; (**C**) Tbr batch #C (magnification ×20K).

**Figure 7 pharmaceutics-13-01541-f007:**
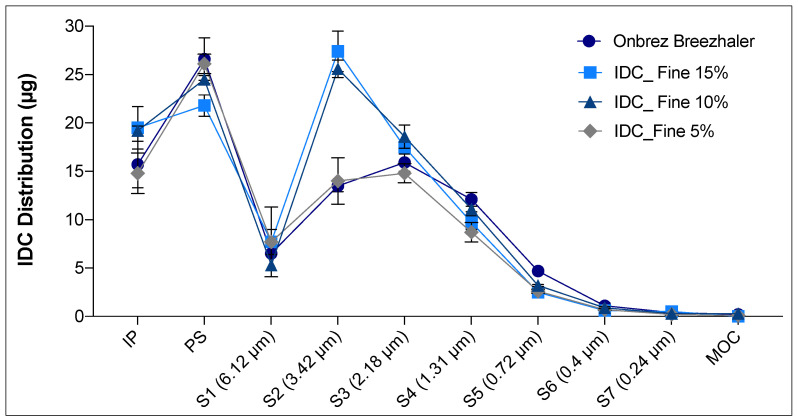
Indacaterol DPI (150 µg/25 mg of blend) distribution in the NGI after a single dose aerosolisation. Three batches containing different amounts of fine lactose (IDC_Fine 5%, IDC_Fine 10%, IDC_Fine 15%) were compared with Onbrez (*n* = 3, mean ± SD). Upper cut-off diameter of stages (S) is reported in the brackets, (IP = induction port, PS = pre-separator and MOC = micro orifice collector).

**Figure 8 pharmaceutics-13-01541-f008:**
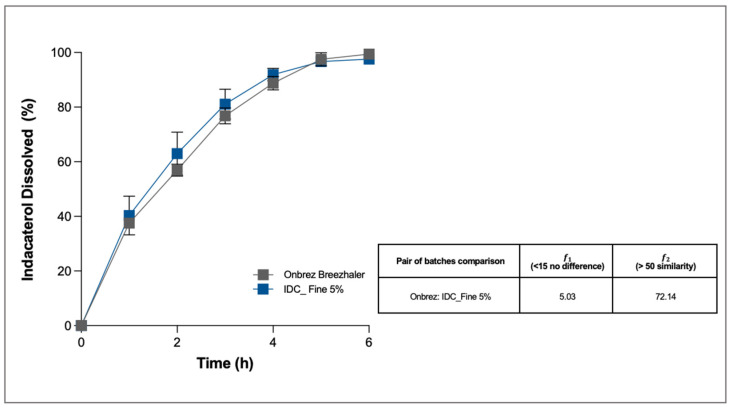
Cumulative mass (%) dissolution profiles of indacaterol test formulation with 5% of fine lactose (IDC_Fine5%) in comparison with the reference product, Onbrez, after aerosolisation (*n* = 3, mean ± SD). Difference (𝑓1) and similarity factor (𝑓2) between the two profiles were calculated.

**Figure 9 pharmaceutics-13-01541-f009:**
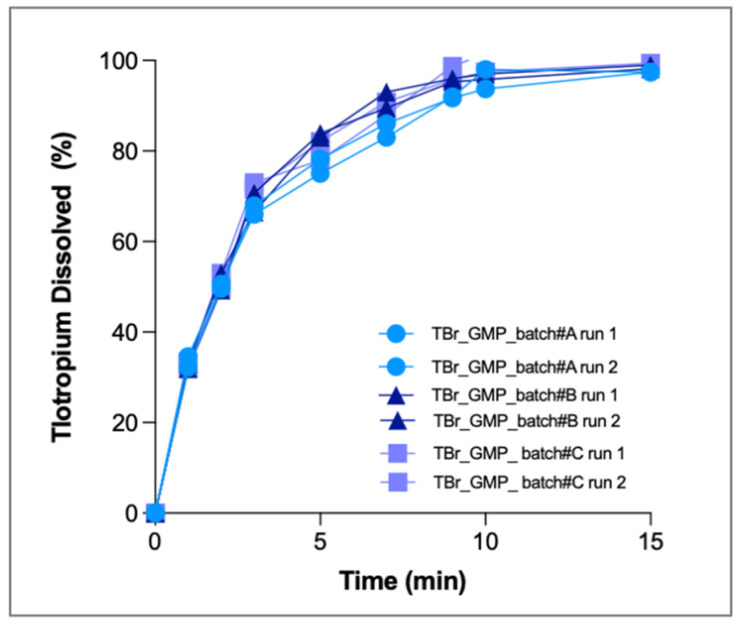
Cumulative mass (%) dissolution profiles of tiotropium bromide released by three small-scale GMP batches after aerosolisation.

**Table 1 pharmaceutics-13-01541-t001:** Indacaterol (IDC) µg remained inside the capsule and inhaler after aerosolization of the reference product Onbrez and tested formulations containing different amount of fine lactose (*n* = 3, mean ± SD).

IDC (µg)	Onbrez Breezhaler	IDC_Fine 5%,	IDC_Fine 10%,	IDC_Fine 15%
Emitted Dose	97.8 ± 1.1	95.6 ± 1.2	109.0 ± 0.4	107.2 ± 1.2
Capsule and Inhaler	24.9 ± 6.3	36.4 ± 0.9	36.8 ± 0.5	34.8 ± 0.7

## Data Availability

The data presented in this study are available on request from the corresponding author.

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
