# Peer review of "RespiCellTM: An Innovative Dissolution Apparatus for Inhaled Products"

_pharmaceutics, 2021, doi:10.3390/pharmaceutics13101541_

Round 1
Reviewer 1 Report
Major comments
This manuscript focused on the use of RespiCell in a testing the dissolution of a range of inhalable powders. The authors claim that this apparatus is superior to other methods of dissolution tests for pharmaceutical aerosols. However, the design of RespiCell appears to be essentially the same as that of the Franz cell, according to Figure 1 and Section 2.4.1. How is the former different or superior to the latter? Powders can also be dispersed onto a filter membrane in an impactor and then placed on a Franz cell for the same type of dissolution experiments. The authors should explain the novelty of RespiCell and justify its use over the Franz cell. The English needs to be improved. There are grammatical errors throughout the text. Some of the terms or expressions used seem odd. A few abbreviations were not spelt out in full at their first appearance.
Specific comments
- Line 20: The words “in vitro” should be italicized. This should be done throughout the text for both “in vitro” and “in vivo”. “API” should be spelt out in full in the abstract.
- Line 30: The words “behaviour” and “aerosolization” are in British and American spelling, respectively, whereas “behaviors” in Line 137 is American. The spelling style should be kept consistent.
- Line 33: Change “Respicell” to “RespiCell”.
- Line 110: PK should be spelt out in full at its first appearance.
- Line 130: Change “non dissolution rate limiting” to “non-dissolution rate-limiting”.
- Line 136: PD should be spelt out in full at its first appearance.
- Line 137: Change “investigation the dissolution” to “investigation of the dissolution”.
- Line 138: Change “APIs” to “API”.
- Section 2.1: Provide the grade and manufacturer’s details for lactose carrier. Was it lactose monohydrate or lactose anhydrous?
- Line 149: Change “purchased by” to “purchased from”. Magnesium stearate should not be capitalised.
- Lines 161-162: The composition of the SLF is missing in the supplementary material.
- Line 163: Spell out SLF at its first appearance.
- Line 176: Change “methods” to “method”.
- Line 182: The injection volume, column temperature, and the gradient elution schedule should be included for the HPLC methods in the supplementary material.
- Line 190: Magnesium stearate was already abbreviated in Line 149 so only the abbreviation is needed here. Change “blended and with” to “blended with”.
- Line 192: Change “was of 20 g” to “was 20 g”.
- Line 199: Indacaterol maleate should not be capitalised.
- Line 215: Change “media” to “medium”.
- Line 239: How much dissolution medium is filled into the receptor chamber for each experiment?
- Line 241: How much sample is withdrawn at each timepoint?
- Section 2.4.2: Powder dispersion could be affected by the temperature and relative humidity of the environment. Please specify those values and how they were controlled in the impactor runs.
- Line 261: Change “activation” to “sampling”.
- Section 2.4.3: The composition of the dissolution media for the various powder formulations were different. The authors should explain the rationale for how they chose the dissolution media. Why did they use different surfactants at different concentrations? The duration and sampling timepoints of the dissolution runs should be specified.
- Line 274: What was the pH of the PBS?
- Line 282: What was the pH of the SLF?
- The numbering of Sections 2.2.6 and 2.2.7 do not follow that of previous sections. The sections before 2.2.6 are 2.4.1, 2.4.2, and 2.4.3.
- Line 381: What is meant by “without undergoing any metallization process”? Does it mean that no metallic coating was applied onto the samples? If so, how did the authors avoid charging and subsequent deformation of the particles when the samples were viewed under the SEM?
- Figure 2: The markers of the three dissolution profiles should be joined by lines to improve clarity. The logical order of the powders in the legend should be raw tobramycin, MgSt 0.5%w/w, and 5% w/w MgSt.
- Lines 392-393: The method of determining dose accuracy should be included in Materials and Methods.
- Line 496: Change “BE” to bioequivalence.
- Line 535: It would be helpful to have a table showing the aqueous solubility of all the drugs used in the study so that the readers can have a clearer idea on the differences between their solubilities. Indacaterol maleate was already mentioned earlier so the abbreviation should start at its first appearance rather than here.
- Line 540: The cascade impaction study was never mentioned in the Materials and Methods section. The details of it should be stated there. From Figure 7, the impactor was presumably the Next Generation Impactor. If so, please show the whole deposition profile including the amount of drug in the capsule, inhaler, adaptor, throat, and pre-separator.
- Line 544: Change “Figure 5” to “Figure 7”.
- Line 545: Change “superior” to “higher”.
- Line 564: Change “superimpose” to “superimposable”.
- Line 602: Spell out “CQA” in full.
- Line 621: Change “to e” to “to the”.
Reviewer 2 Report
The current manuscript aimed to focus on feasibility of custom-built dissolution apparatus for inhalable products. This manuscript brings an important quality control tool to understand in vitro in vivo relationship of the inhaled products. The authors have tested various drugs (hydro and hydrophilic) and blends of drug plus excipients to understand how their custom dissolution apparatus can predict drug release. Most importantly, the lab-based inhalable pilot formulations were compared to the industrial batch and the efficiency of the novel apparatus was found to be excellent.
The experiment is well designed, proposes novel concepts and data is well analysed. The manuscript is suitable for publication in Pharmaceutics after addressing the following comments.
Comments:
- Authors have proposed a novel apparatus called RespiCell and focused on defining its merit with a range of examples. I am wondering if the authors have compared the merits of using RespiCell vs other published dissolution tools for pulmonary products.
- It makes sense to conduct dissolution studies with only relevant sized particles. However, in practical situations it is impossible to generate such as uniform size particles and the particles size that is being dosed remains highly variable. I am surprised why authors have excluded the dissolution of whole particles not just by selecting particles less than 5-micron size from the impactor. It would be clear to readers if both were compared.
- I am concerned with dissolution media. Authors are suggested to describe the rationale for selecting the dissolution media for both hydrophilic and hydrophobic drugs. Why more bio relevant media were not selected?
- Introduction is too long which can be trimmed to make it short.
- Figures can be combined where appropriate.
- Authors are requested to correct the following typos and sentence structure.
Line 72: please mention lungs endogenous lung surfactant
Line 73: Please replace lower and upper lung with “lower and upper respiratory tract”
Line 75: Add a sentence about particle clearance mechanism in the lower respiratory tract
Line 110: change “man” to “human”
Line 326-328. Please change the writing style. It is confusing.
Line 335: change “ analysys” to analysis
Round 2
Reviewer 1 Report
The manuscript is much improved and most comments have been addressed. The only change needed is that the amount of drug deposits in the capsule, inhaler, adaptor, throat, and pre-separator should be shown in Figure 7. It is stated in Section 2.6 that the amount of drug on these parts were quantified but the data are not shown. The deposition profiles are thus incomplete. Please show the whole set of data in Figure 7 by including the drug deposits on those parts.
